# Heteropolyacid Ionic Liquid-Based MCF: An Efficient Heterogeneous Catalyst for Oxidative Desulfurization of Fuel

**DOI:** 10.3390/ma16083195

**Published:** 2023-04-18

**Authors:** Tingting Pei, Yaxian Chen, Huiting Wang, Lixin Xia

**Affiliations:** 1College of Chemistry, Liaoning University, Shenyang 110036, China; 18241353285@163.com (T.P.); cyx512148@163.com (Y.C.); 2Liaoning Key Laboratory of Chemical Additive Synthesis and Separation, Yingkou Institute of Technology, Yingkou 115014, China

**Keywords:** heteropolyacids, ionic liquids, mesostructured cellular silica foam, oxidative desulfurization

## Abstract

A new type of catalyst was synthesized by immobilizing heteropolyacid on ionic liquid-modified mesostructured cellular silica foam (denoted as MCF) and applied to the oxidative desulfurization of fuel. The surface morphology and structure of the catalyst were characterized by XRD, TEM, N_2_ adsorption–desorption, FT-IR, EDS and XPS analysis. The catalyst exhibited good stability and desulfurization for various sulfur-containing compounds in oxidative desulfurization. Heteropolyacid ionic liquid-based MCF solved the shortage of the amount of ionic liquid and difficult separation in the process of oxidative desulfurization. Meanwhile, MCF had a special three-dimensional structure that was not only highly conducive to mass transfer but also greatly increased catalytic active sites and significantly improved catalytic efficiency. Accordingly, the prepared catalyst of 1-butyl-3-methyl imidazolium phosphomolybdic acid-based MCF (denoted as [BMIM]_3_PMo_12_O_40_-based MCF) exhibited high desulfurization activity in an oxidative desulfurization system. The removal of dibenzothiophene could achieve levels of 100% in 90 min. Additionally, four sulfur-containing compounds could be removed completely under mild conditions. Due to the stability of the structure, sulfur removal efficiency still reached 99.8% after the catalyst was recycled six times.

## 1. Introduction

With the increasing demand for fuel consumption worldwide, environmental legislation implemented by various countries is increasingly strict, including the requirements for sulfur content [1,2]. Therefore, the study of ultra-deep desulfurization attracted increasing research attention [3,4,5] and many efficient desulfurization techniques have been developed. These include hydrodesulfurization (HDS) [6,7,8], extractive desulfurization (EDS) [9,10,11], adsorption desulfurization (ADS) [12,13,14,15] and oxidative desulfurization [16,17,18,19]. Hydrodesulfurization technologies are effective on most sulfides but have obvious disadvantages such as high temperature and pressure, and difficulty in removing aromatic sulfur compounds. These shortcomings can be solved by oxidative desulfurization technologies. Oxidative desulfurization not only requires mild operating conditions but is also effective in the removal of aromatic sulfur compounds. Oxidative desulfurization is regarded as a green and efficient deep desulfurization technology and has great development prospects.

High-efficiency catalysts are the key factor affecting the efficiency of the removal of sulfur compounds. Accordingly, increasing research has been focused on catalyst development. To date, catalysts widely used in oxidative desulfurization include polyoxometalates [20,21,22,23,24], ionic liquids [25,26,27,28], organic acid [29,30,31] and carbon materials [32,33]. In particular, polyoxometalates have received increasing attention as functional catalysts in organic reaction systems [34,35]. Due to their unique properties, polyoxometalates have advantages including good selectivity, high hydrothermal stability, oxidation–reduction ability, and strong acidity in oxidative desulfurization. However, due to the small specific surface area of heteropolyacids, which is not conducive to promoting catalytic reactions, their catalytic performance is greatly limited; this affects their application value in catalysis. Therefore, to address this shortcoming, it is of great significance to design and synthesize catalysts with more catalytic active sites by selecting a suitable support with a higher specific surface area.

Homogeneous catalysts usually show high catalytic activity in oxidative desulfurization but their separation and regeneration are difficult. Compared with homogeneous catalysts, heterogeneous catalysts have the advantages including ease of separation and regeneration but the catalytic active sites are less exposed to the reactants. Therefore, there is an urgent need to develop a method for preparing efficient catalysts for oxidative desulfurization that can not only be readily separated and regenerated but also have the high catalytic activity of heterogeneous catalysts.

Mesostructured cellular silica foam (MCF) [36,37,38] can be widely used in sensor, separation, adsorption, electronic insulation and other fields due to its large specific surface area, ordered pore structure and high hydrothermal stability. In contrast to other mesoporous materials, MCF is a kind of mesoporous silica with an open three-dimensional pore structure and large pore size, so it is often regarded as an excellent support. MCF as a heterogeneous catalyst support greatly increases the surface area, providing many sites for the reaction to take place. However, MCF has some drawbacks as a catalyst: the acid concentration on its surface is low and the acidity is weak. Furthermore, the weak interaction between active species and MCF can readily lead to the shedding of active substances.

In this study, to overcome these problems, heteropolyacid ionic liquids supported on MCF were designed and synthesized to improve mass transfer and increase catalytic activity sites. The resulting heteropolyacid ionic liquid-modified MCF catalyst was applied to the extractive and catalytic oxidative desulfurization (ECODS) system, wherein it proved to have the dual functional advantages of MCF and the heteropolyacid ionic liquids. The catalyst showed excellent desulfurization performance, could be recycled many times and was easy to recover and reuse. The excellent performance of the catalyst makes it very promising for industrial applications, across multiple fields.

## 2. Materials and Methods

### 2.1. Chemicals

Silicotungstic acid hydrate (H_4_SiW_12_O_40_), phosphomolybdic acid hydrate (H_3_PMo_12_O_40_), phosphotungstic acid hydrate (H_3_PW_12_O_40_), 1-methylimidazole, (3-chloropropyl) trimethoxysilane, *n*-octane, hydrochloric acid, 30 wt% hydrogen peroxide, thiophene (T) and toluene were provided by Sinopharm Chemical Reagent Co., Ltd. and were of analytical grade. Poly(ethylene glycol)-block-poly(propylene glycol)-block-poly(ethylene glycol) (P123), benzothiophene (BT), 4,6-dimethyldibenzothiophene (4,6-DMDBT) and dibenzothiophene (DBT) were obtained from Sigma-Aldrich. 1,3,5-trimethylbenzene (TMB) and ethyl silicate (TEOS) were purchased from Aladdin Chemistry. Ionic liquid [BMIM]BF_4_ was synthesized according to a published procedure [39,40].

### 2.2. Catalyst Preparation

#### 2.2.1. Preparation of MCF

MCF was synthesized according to a published procedure [41]. P123 was dissolved in HCl (1.6 M, 150 mL) with mechanical stirring for 2 h. Then, TMB was added to the solution and the resulting mixture was stirred at 38 °C for 1 h. Subsequently, TEOS was gradually added to the flask whilst stirring at 38 °C for 20 h. The mixture was heated in a hydrothermal reactor at 110 °C for 24 h and washed with toluene. The precipitate was calcined at 500 °C for 8 h. After removing the template, a dry white solid power was obtained.

#### 2.2.2. Grafting of 1-Butyl-3-methylimidazolium Chloride ([BMIM]Cl) on MCF

MCF (2.0 g) and (3-chloropropyl) trimethoxysilane (8 mL) were heated at reflux with toluene under the protection of nitrogen at 130 °C for 24 h. The obtained solution was filtered and washed with toluene several times. The precipitate (2.0 g) and 1-methylimidazole (8 mL) were refluxed for 24 h. The cooled mixture was isolated by filtration and dried for 24 h under vacuum. Finally, the [BMIM]Cl-based MCF product was obtained as a white power [39].

#### 2.2.3. Preparation of Heteropolyacid Ionic Liquid-Based MCF

The catalyst of heteropolyacid ionic liquid-based MCF were prepared through the anion exchange reaction of heteropolyacids (HPAs) with [BMIM]Cl-based MCF. Taking [BMIM]_3_PW_12_O_40_-based MCF as an example, 20 mg of [BMIM]Cl-based MCF was ultrasonically dispersed into a saturated solution of phosphotungstic acid hydrate (30 mL) with stirring for 48 h. The precipitate was isolated by filtration and washed four times with deionized water to remove excess phosphotungstic acid. The product of [BMIM]_3_PW_12_O_40_-based MCF was acquired after drying in the vacuum drying oven. In the same way, [BMIM]_4_SiW_12_O_40_-based MCF and [BMIM]_3_PMo_12_O_40_-based MCF were prepared [42,43].

#### 2.2.4. Catalyst Characterization

The surface morphology of the ionic liquid-based MCF was analyzed by transmission electron microscopy (TEM; JEM-2100; JEOL HITACHI; Japan) at 100 kV. Powder X-ray diffraction (XRD; Bruker; Germany) analysis was carried out on a Bruker instrument with high-intensity Cu Ka radiation (λ = 1.54 Å). N_2_ adsorption–desorption isotherms were investigated by surface area analyzer. FTIR spectrophotometry (PerkinElmer; USA) was performed using a KBr disc. Surface element analysis was performed by X-ray photoelectron spectroscopy (XPS; Thermo Scientific ESCALAB250; Waltham; USA). The removal of sulfur-containing compounds were determined using high-performance liquid chromatography (HPLC; LC-20A Prominence; Japan). The mobile phases of HPLC was methanol in water. The parameters of 4,6-DMDBT, DBT, BT and T were set as 90%, 90%, 80% and 70% with a flow rate of 1 mL/min, respectively.

#### 2.2.5. Preparation of Model Oil and Desulfurization

The model oil containing DBT was made up by dissolving DBT (0.588 g) in *n*-butane (100 mL), giving with a corresponding sulfur concentration of 1000 ppm. The model oils of T, BT, and 4, 6-DBT were prepared using the same method with *n*-octane as the solvent. 

Taking [BMIM]_3_PW_12_O_40_-based MCF as an example, each desulfurization experiment was conducted in a 10 mL round bottom flask, to which ionic liquid [BMIM]BF_4_ (3 mL) and a portion of the catalyst were added and dispersed uniformly at room temperature by stirring for 40 min. Then, hydrogen peroxide (30 wt%) and model oil (5 mL) were added, sequentially. The desulfurization experiment was carried out in an oil bath at the reaction temperature of the extractive catalytic oxidation. While the experiment was in process, the upper phase containing the model oil was sampled and the sulfur concentration was measured by HPLC every 30 min. 

At the end of one desulfurization experiment, the upper oil phase was separated by decantation. The lower ionic liquid and catalyst were washed with diethyl ether and dried for reuse, while fresh hydrogen peroxide oxidant and model oil were injected for the next run of desulfurization experiment.

## 3. Results and Discussion

### 3.1. Morphological Characterization of Heteropolyacid Ionic Liquids-Based MCF

The wide-angle XRD pattern of the MCF and heteropolyacid ionic liquid-based MCF is shown in Figure 1. The results showed that the [BMIM]_3_PMo_12_O_40_-based MCF and [BMIM]_3_PW_12_O_40_-based MCF had corresponding characteristic peaks of heteropolyacids. Notable peaks corresponding to phosphomolybdic acid and phosphotungstic acid were also observed in the wide-angle XRD patterns, indicating that the heteropolyacids were successfully supported on MCF.

The surface morphology and structure of MCF modified with different heteropolyacid ionic liquids were characterized by TEM. As shown in Figure 2a, the TEM image of MCF exhibited a honeycomb foam structure with a disordered pore structure, a pore size ratio of approximately 20–30 nm and a pore wall thickness of approximately 2 nm, which are typical of mesoporous materials. The morphologies of the [BMIM]Cl-based MCF materials following the anion-exchange reaction are shown in Figure 2b–d. A comparison of the TEM images of the [BMIM]_3_PMo_12_O_40_-based MCF, [BMIM]_3_PW_12_O_40_-based MCF and [BMIM]_4_SiW_12_O_40_-based MCF revealed that the morphology of MCF modified with different heteropolyacid ionic liquids did not change significantly. The honeycomb foam mesostructure of MCF was still preserved after the introduction of the heteropolyacids, indicating that the introduction of the catalytic anion did not damage the mesoporous material structure.

Figure 3 shows the N_2_ adsorption–desorption isotherms of MCF and the heteropolyacid ionic liquid-based MCF. All adsorption–desorption isotherms showed a typical IV-type curve with an H1-type hysteresis ring, which is characteristic of mesoporous samples. With the introduction of the heteropolyacid ionic liquids, the capillary condensation height decreased, indicating that the heteropolyacid ionic liquids were successfully supported on MCF. The BET surface areas of MCF and the heteropolyacid ionic liquid-based MCF samples were 510.78 and 365.83 m^2^·g^−1^. The decrease in the BET surface area suggested that the heteropolyacid ionic liquids were successfully grafted onto MCF.

### 3.2. Composition and Elemental Analysis of Heteropolyacid Ionic Liquid-Based MCF

The different fabrication of heteropolyacid ionic liquid-modified MCF obtained by the ionic exchange reaction between the [BMIM]Cl-based MCF and HPA were characterized by FTIR spectroscopy (Figure 4), the image on the right is an enlarged image of the shadow part on the left. As shown in the FTIR spectrum of MCF (curve a), the absorption bands located at 1081 and 958 cm^−1^ were attributed to the Si-O-Si stretching vibrations of MCF, while the characteristic absorption peaks located at 469 and 791 cm^−1^ were attributed to the bending vibrations and stretching vibrations of Si-OH. Additionally, the C-H characteristic absorption peak of the imidazole ring band was observed at 1150 cm^−1^ (curve b). Similar absorption bands were observed for the other heteropolyacid ionic liquid-based MCF compounds, as shown in curves c–e. Moreover, the characteristic bands of the appearance at 1068, 791 and 971 cm^−1^ in curve c were attributed to vibrations of P-O, Mo-Oe-Mo and Mo=O in the [BMIM]_3_PMo_12_O_40_-based MCF. Furthermore, the bands at 1081, 982 and 807 cm^−1^ in curve d were attributed to characteristic absorption peaks caused by vibrations of P-O, W=O and W-Oe-W in the [BMIM]_3_PW_12_O_40_-based MCF. In curve e, the absorption peak at 1094 cm^−1^ was attributed to Si-O stretching vibrations, while the band at 978 cm^−1^ was attributed to the anti-symmetric stretching vibrations of W=O and the bands located at 926 and 804 cm^−1^ were attributed to the absorption peak caused by the anti-symmetric stretching vibrations of W-O-W [44]. Based on the above analysis, it was concluded that the presence of heteropolyacid anion groups confirmed the heteropolyacids were successfully supported on [BMIM]Cl-based MCF.

The chemical composition and state of the catalysts obtained by the reaction between heteropolyacid and [BMIM]Cl-based MCF were further investigated by XPS analysis. The XPS spectrum (Figure 5a) showed characteristic absorption peaks at binding energies of 102.4, 197.4, 285.5, 399.2 and 532.9 eV, which corresponded to the presence of imidazolinyl functional groups of Si 2p, Cl 2p, C 1s, N 1s and O 1s, respectively. These typical characteristic peaks revealed the success of the ionic liquid process could be confirmed via the Cl 2p electrons appeared at 197.4 eV.

As shown in the XPS, the presence of characteristic peaks of heteropolyacid ionic liquid-based MCF in Figure 5b–d. It demonstrated the catalysts of [BMIM]_3_PMo_12_O_40_-based MCF were composed of the elements of Si, O, C, N, P and Mo. It can be seen the absence of the characteristic peak for Cl 2p located at 197.4 eV, while the appearance of new peaks of P 2p, Mo 3d in XPS wide scans. Similarly, the characteristic peaks of [PW_12_O_40_]^3−^ and [SiW_12_O_40_]^4−^ were observed clearly. The experimental results agreed with the energy spectrum and further confirmed the successful preparation of heteropolyacid ionic liquid-based MCF by anion-exchange reaction.

Elemental analyses of various catalysts related to [BMIM]Cl-based MCF, [BMIM]_3_PMo_12_O_40_-based MCF, [BMIM]_3_PW_12_O_40_-based MCF and [BMIM]_4_SiW_12_O_40_-based MCF were performed using energy-dispersive spectroscopy (EDS). As shown in Figure 6, the different ionic liquid-modified MCF compounds contained different characteristic elements. The EDS spectrum of the [BMIM]Cl-based MCF not only contained the characteristic peaks of the O and Si elements of MCF, but also the characteristic peaks of Cl Kα, Cl Kβ, Cl L and N were also observed. This result indicated that functionalization with catalytic anions was successfully achieved by heteropolyacid ionic liquid immobilization on MCF. Such a phenomenon was evident by the presence of binding energies of 1.72, 2.02 and 2.40 KeV, which corresponded to the characteristic peaks of W, P and Mo, respectively. When the anion-exchange reaction occurred between [BMIM]Cl-based MCF and heteropolyacid, it proved that the heteropolyacid ionic liquids were successfully modified with the MCF that laid the foundation on the next step of catalytic oxidative desulfurization.

### 3.3. Influence of Different Catalyst on Sulfur Removal of DBT

The catalytic oxidative desulfurization activity of MCF modified with different heteropolyacid ionic liquids was compared under the same desulfurization reaction conditions for the removal of DBT (Figure 7). The desulfurization efficiency of three kinds of catalysts supported by different heteropolyacids increased over time. The desulfurization efficiency with DBT increased at different rates until equilibrium was reached. The negative charge of heteropolyanion depended first on electron reducibility. Compared with [PMo_12_O_40_]^3−^ and [PW_12_O_40_]^3−^, Keggin heteropolyacid catalysts with [SiW_12_O_40_]^4−^ had a worse result in reducibility due to a smaller capacity to accept electrons [45]. Due to the impact of heteroatom substitution on oxidizing power, the catalytic desulfurization efficiency of the [BMIM]_3_PMo_12_O_40_-modified MCF and [BMIM]_3_PW_12_O_40_-modified MCF was much higher than [BMIM]_4_SiW_12_O_40_-modified MCF. With the increase in time to 1.5 h, the desulfurization efficiency of DBT catalyzed by [BMIM]_3_PW_12_O_40_-modified MCF, [BMIM]_3_PMo_12_O_40_-modified MCF and [BMIM]_4_SiW_12_O_40_-modified MCF reached 75.24%, 94.56% and 56.01%, respectively. As the reaction continued, 100% sulfur removal conversion was first achieved using [BMIM]_3_PMo_12_O_40_-modified MCF. Under the same experimental conditions, the sequence of catalytic efficiency of different acid catalysts was as follows: [BMIM]_3_PMo_12_O_40_-based MCF > [BMIM]_3_PW_12_O_40_-based MCF > [BMIM]_4_SiW_12_O_40_-based MCF. Therefore, it was speculated that the presence of different heteropolyanions in the catalysts resulted in different catalytic desulfurization activity. Based on these results, it was determined that the [BMIM]_3_PMo_12_O_40_-based MCF had better sulfur removal performance on DBT during the catalytic desulfurization process and was regarded as the proper catalyst for subsequent experimentation.

### 3.4. Effect of the Amount of [BMIM]_3_PMo_12_O_40_-Based MCF on Sulfur Removal of DBT

Figure 8 shows the effect of the amount of catalyst on sulfur removal efficiency, using DBT. It can be seen that, under the same conditions, sulfur removal was continuously promoted, accompanying the catalyst dosage increase from 15 to 40 mg within 3 h. It was evident that the number of catalytic sites contributed to the increase in catalyst dosage, leading to significant positive impacts on the desulfurization efficiency of the system. When the desulfurization reaction was carried out for 2 h with 30 mg, the DBT in the model oil was completely removed. There was no significant difference in the desulfurization performance at 30 and 40 mg dosages. Thus, to optimize material efficiency, the optimal dosage amount of [BMIM]_3_PMo_12_O_40_-based MCF was determined to be 30 mg.

### 3.5. Influence of O/S Ratio on Sulfur Removal

Due to its superiority, hydrogen peroxide plays an important role as an oxidant in the process of oxidative desulfurization. Stoichiometrically, 2 mol of hydrogen peroxide is required for the oxidation of sulfur components to the corresponding sulfones [46]. However, a higher oxygen/sulfur ratio (O/S) is needed in practice because of the competition between the oxidation of DBT and the self-decomposition reaction of hydrogen peroxide. Accordingly, Figure 9 shows the effect of the O/S ratio on sulfur removal using DBT. When the molar ratio of O/S increased from 2 to 6, sulfur removal efficiency markedly increased from 78% to 100% in 1.5 h. There was no evident change in sulfur removal as the molar ratio H_2_O_2_/DBT gradually increased up to 10. These results revealed that the amount of oxidant required in the system had reached saturation at an O/S ratio of 6. Therefore, considering the reality of the reaction system, a molar ratio H_2_O_2_/DBT of 6 was regarded as the most suitable value in the oxidative desulfurization system. 

### 3.6. Sulfur Removal of Different Temperatures

The influence of temperature on the desulfurization efficiency is shown in Figure 10. The experimental conditions were as follows: O/S = 6, V_DBT_ = 5 mL, V_IL_ = 3 mL, m = 30 mg, and t = 180 min. When the reaction temperature increased from 20 to 60 °C, the rate of desulfurization increased markedly from 69.3% to 100% within 2 h. Under the same experimental conditions, 60 °C was the lowest temperature at which a sulfur removal rate of 100% was achieved. Higher temperatures promote the desulfurization process but higher temperatures mean higher costs of production. Accordingly, 60 °C was selected as the optimal reaction temperature for sulfur removal using DBT under the experimental conditions. 

### 3.7. Sulfur Removal of Different Sulfur-Containing Compounds

Different sulfur-containing compounds T, BT, DBT and 4,6-DMDBT were used in the model oil as representatives to investigate the desulfurization efficiency of the ECODS system under the optimal experimental conditions. As the desulfurization process progressed, the sulfur content of the different model oils gradually decreased. The removal of T eventually was up to 74% within 3 h, while the removal of BT, DBT and 4,6-DMDBT reached 99.63%, 100% and 99.75%, respectively. As can be seen from Figure 11, sulfur removal efficiency decreased as follows: DBT > 4,6-DMDBT > BT > T. At the same time, the electron density and steric hindrance of the sulfur atom were two critical factors for desulfurization efficiency. It is known that a higher electron density and lower steric hindrance contribute to a higher desulfurization efficiency in desulfurization systems. The electron densities of the sulfur atoms in 4,6-DMDBT, DBT, BT and T are 5.760, 5.758, 5.739 and 5.696, respectively; thus, the desulfurization rates using T, BT and DBT were affected by the electron density [47]. However, 4,6-DMDBT was unfavorable for its interactions with the catalytic centers because of the steric hindrance imposed by the methyl group, with the effect that the sulfur removal rate of 4,6-DMDBT was lower than that of DBT.

### 3.8. Sulfur Removal of Different Desulfurization Systems

After optimizing the experimental conditions, sulfur removal of different desulfurization systems was compared in Figure 12. As shown in curve b, when the desulfurization reaction was carried out for 120 min, the individual extractive desulfurization efficiency of pure ionic liquid [BMIM]BF_4_ without any catalyst was only 28.11%. As shown in curve c, when [BMIM]Cl-based MCF as a non-catalytic carrier was added to ionic liquid [BMIM]BF_4_, the desulfurization efficiency was 33.89%. This result was similar to the extractive desulfurization system alone, indicating that [BMIM]Cl-based MCF had no catalytic function to improve the sulfur removal rate. For the extractive and catalytic oxidative desulfurization (ECODS) model, pure H_3_PMo_12_O_40_ was used as the catalyst with [BMIM]BF_4_ as the extractant (curve d), resulting in a sulfur removal rate of 72.6%. This indicated that the desulfurization efficiency was promoted in the pure acidic catalyst system. However, as it can be seen from curve e with the [BMIM]_3_PMo_12_O_40_-based MCF as the catalyst, the desulfurization rate was 100% within 90 min. Therefore, under the optimal experimental conditions, the ECODS system with [BMIM]_3_PMo_12_O_40_-based MCF as the catalyst produced the best desulfurization performance. This was ascribed to the dual advantages of [BMIM]_3_PMo_12_O_40_-based MCF—the structural advantages of MCF and the catalytic properties of the heteropolyacids; in tandem, these characteristics increased the contact between the catalytic active sites and the fuel system.

### 3.9. Reusability of [BMIM]_3_PMo_12_O_40_-Based MCF

The reusability and stability of catalysts are significant requirements for its industrial application. Under the optimal experimental conditions, the regeneration of catalysts in the ECODS system was investigated (Figure 13). In the extractive catalytic oxidation process, the sulfide in the oil phase was first extracted by the ionic liquid phase and then the extracted sulfide was further oxidized into the polar sulfone in the presence of [BMIM]_3_PMo_12_O_40_-based MCF and hydrogen peroxide. The first removal and conversion of DBT was completed in fuel. After the first recycling reaction of oxidative desulfurization, the upper oil phase was decanted from the reaction system. The catalyst was isolated from the lower phase by centrifugal separation, then washed with diethyl ether and heated before the next reaction was commenced using fresh model oil and fresh hydrogen peroxide [48]. Sulfur removal efficiency still reached 99.8% after the [BMIM]_3_PMo_12_O_40_-based MCF was recycled six times. Thus, the [BMIM]_3_PMo_12_O_40_-based MCF showed an excellent recycling capability, which was attributed to its structural stability.

## 4. Conclusions

In this paper, MCF modified with heteropolyacid ionic liquids was successfully designed and synthesized. The surface morphology and structure of samples were characterized by XRD, N_2_ adsorption–desorption and TEM, indicating that MCF modified with different heteropolyacid ionic liquids maintains the mesoporous structure of MCF. Because of advantages in structure, heteropolyacid ionic liquid-based MCF was not only highly conducive to mass transfer, but also greatly increased catalytic active sites to improve catalytic efficiency. Simultaneously, heteropolyacid ionic liquid-based MCF solved the difficult separation in the process of oxidative desulfurization. By comparison, [BMIM]_3_PMo_12_O_40_-based MCF exhibited the highest catalytic activity, which also led to the most optimal performance in terms of sulfur removal efficiency in the extractive catalytic oxidation desulfurization system. The removal of dibenzothiophene could achieve levels of 100% in 90 min. Additionally, four sulfur-containing compounds could be removed completely under mild conditions. After the desulfurization process of extractive catalytic oxidation desulfurization system was completed, the regeneration of [BMIM]_3_PMo_12_O_40_-based MCF could be operated by simple separation. In view of [BMIM]_3_PMo_12_O_40_-based MCF showing excellent catalytic activity and stability, it is obvious that the catalyst has a great prospects in terms of industrial application.

## Figures and Tables

**Figure 1 materials-16-03195-f001:**
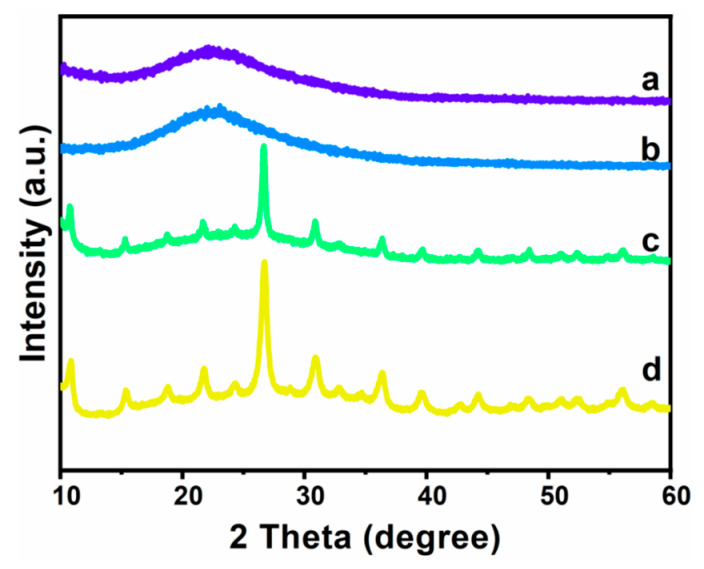
X-ray diffraction patterns of MCF (a), [BMIM]Cl-based MCF (b), [BMIM]_3_PMo_12_O_40_-based MCF (c) and [BMIM]_3_PW_12_O_40-based MCF_ (d).

**Figure 2 materials-16-03195-f002:**
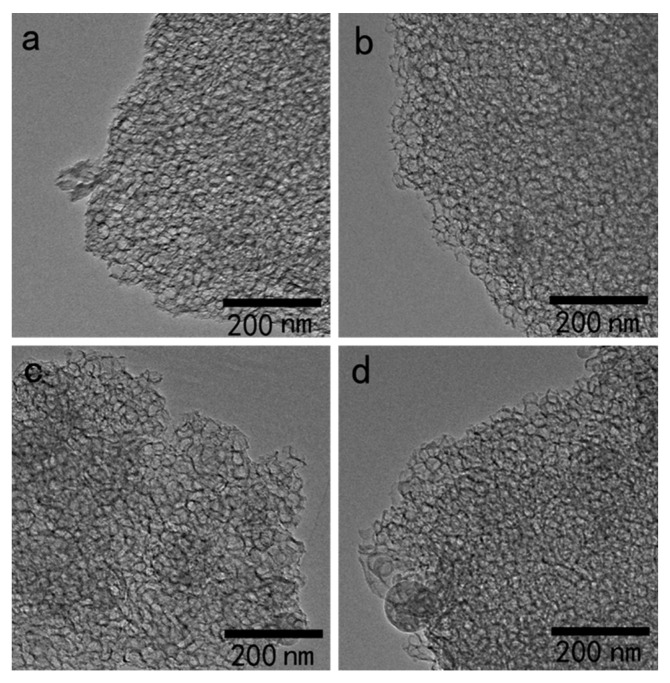
TEM images of MCF (**a**), [BMIM]_3_PMo_12_O_40_-based MCF (**b**), [BMIM]_3_PW_12_O_40_-based MCF (**c**), and [BMIM]_4_SiW_12_O_40_-based MCF (**d**).

**Figure 3 materials-16-03195-f003:**
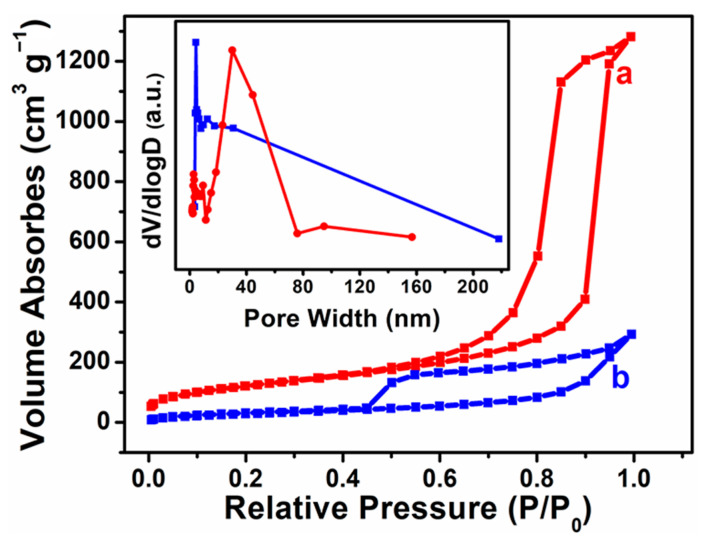
Nitrogen adsorption–desorption isotherms for MCF (a) and heteropolyacid ionic liquid-based MCF (b). Inset: Pore size distribution plots of MCF (red) and heteropolyacid ionic liquid-based MCF (blue).

**Figure 4 materials-16-03195-f004:**
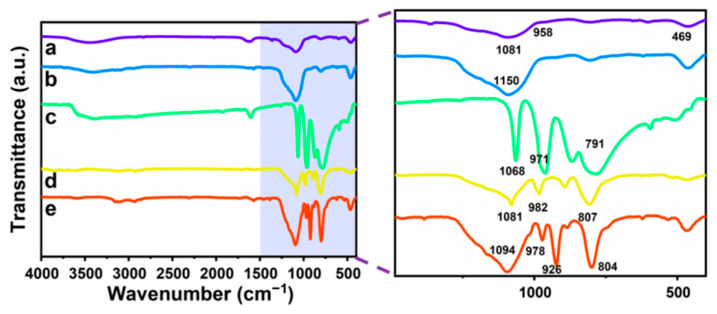
FTIR spectra of MCF (a), [BMIM]Cl-based MCF (b), [BMIM]_3_PMo_12_O_40_-based MCF (c), [BMIM]_3_PW_12_O_40_-based MCF (d), and [BMIM]_4_SiW_12_O_40_-based MCF (e).

**Figure 5 materials-16-03195-f005:**
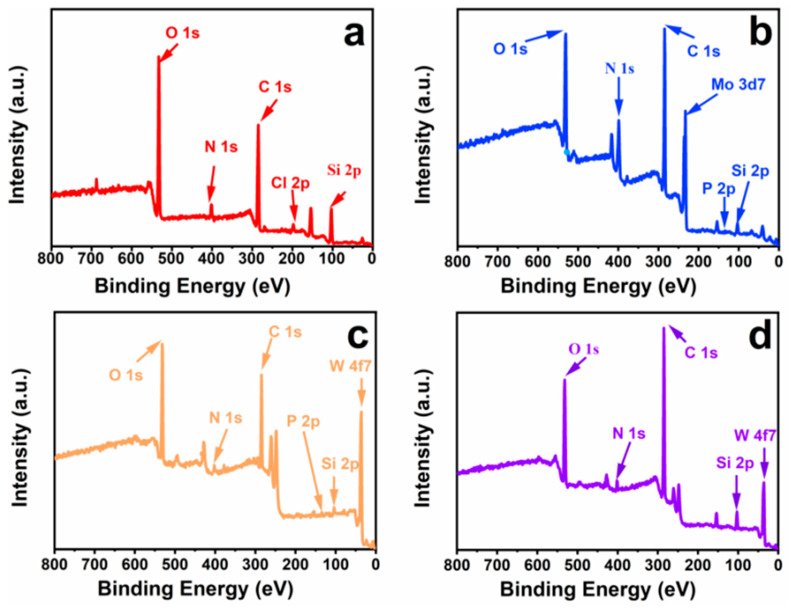
X-ray photoelectron spectroscopy of [BMIM]Cl-based MCF (**a**), [BMIM]_3_PMo_12_O_40_-based MCF (**b**), [BMIM]_3_PW_12_O_40_-based MCF (**c**) and [BMIM]_4_SiW_12_O_40_-based MCF (**d**).

**Figure 6 materials-16-03195-f006:**
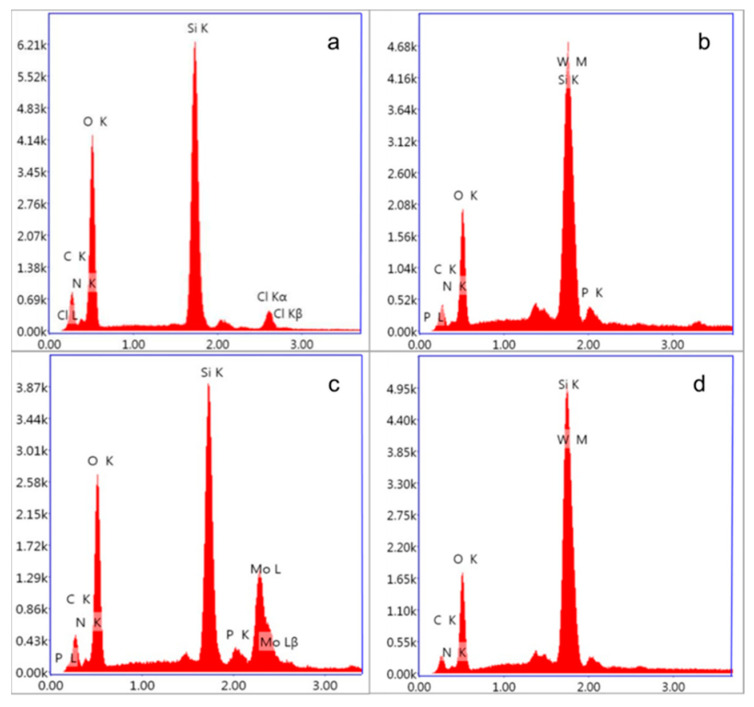
EDS analysis of [BMIM]Cl-based MCF (**a**), [BMIM]_3_PW_12_O_40_-based MCF (**b**), [BMIM]_3_PMo_12_O_40_-based MCF (**c**), and [BMIM]_4_SiW_12_O_40_-based MCF (**d**).

**Figure 7 materials-16-03195-f007:**
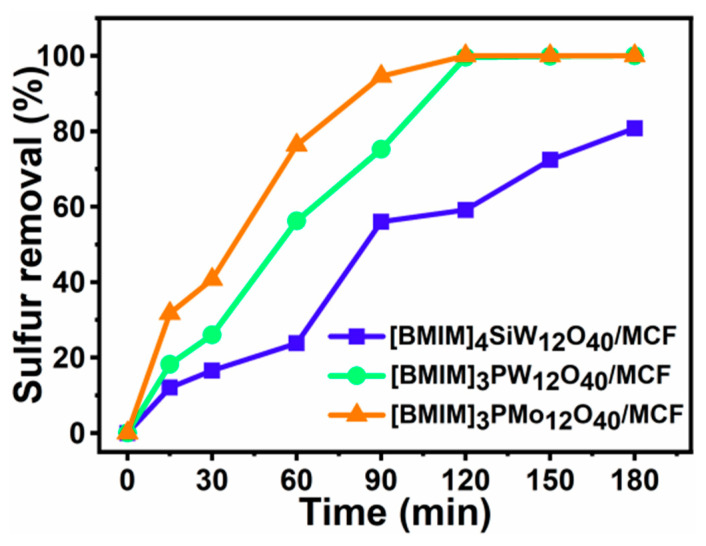
Sulfur removal of DBT with different catalysts. Experimental conditions: T = 60 °C, V_DBT_ = 5 mL, V_IL_ = 3 mL, O/S = 6, and t = 3 h.

**Figure 8 materials-16-03195-f008:**
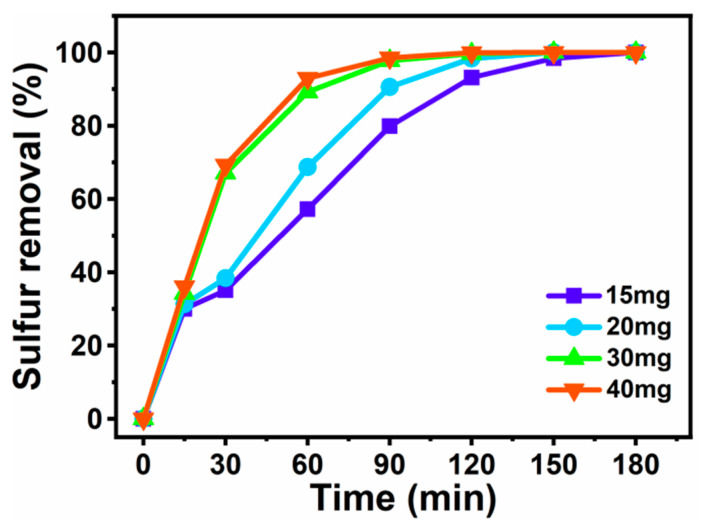
Effect of the amount of [BMIM]_3_PMo_12_O_40-_based MCF on removal of DBT. Experimental conditions: T = 60 °C, V_DBT_ = 5 mL, V_IL_ = 3 mL, O/S = 6, and t = 3 h.

**Figure 9 materials-16-03195-f009:**
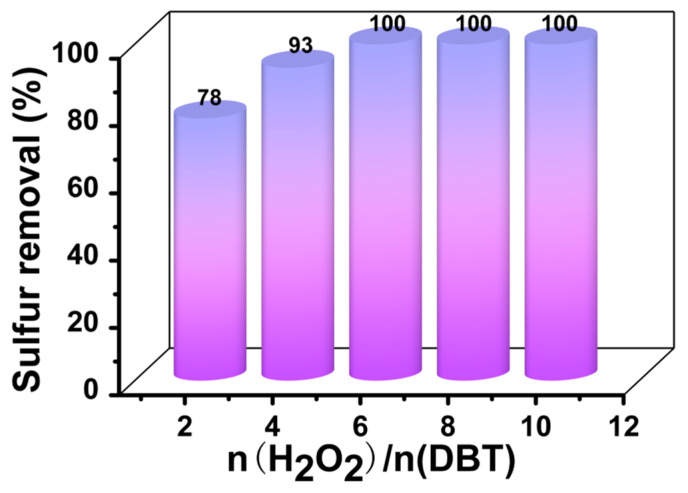
Effect of O/S ratio on sulfur removal efficiency of DBT Experimental conditions: T = 60 °C, V_DBT_ = 5 mL, V_IL_ = 3 mL, m([BMIM]_3_PMo_12_O_40-_based MCF) = 30 mg, and t = 1.5 h.

**Figure 10 materials-16-03195-f010:**
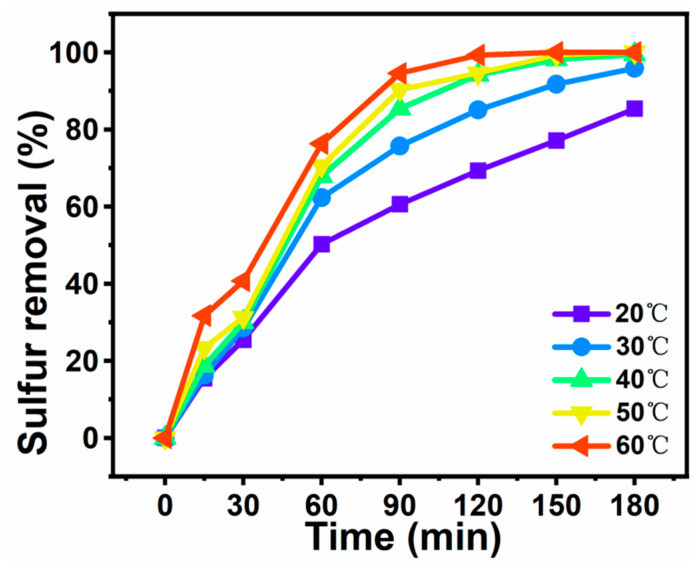
Effects of temperature on sulfur removal efficiency of DBT. Experimental conditions: V_DBT_ = 5 mL, V_IL_ = 3 mL, O/S = 6, m([BMIM]_3_PMo_12_O_40_-based MCF) = 30 mg, and t = 3 h.

**Figure 11 materials-16-03195-f011:**
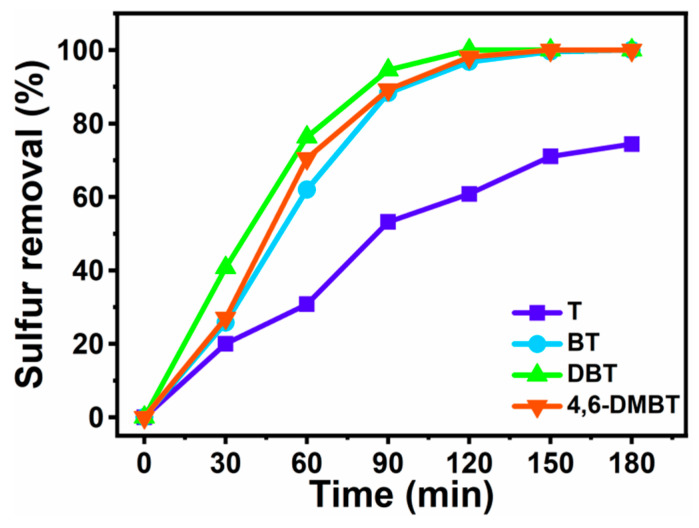
Removal of different sulfur-containing compounds in the desulfurization system. Experimental conditions: T = 60 °C, V_DBT_ = 5 mL, V_IL_ = 3 mL, O/S = 6, m([BMIM]_3_PMo_12_O_40-based MCF_) = 30 mg, and t = 3 h.

**Figure 12 materials-16-03195-f012:**
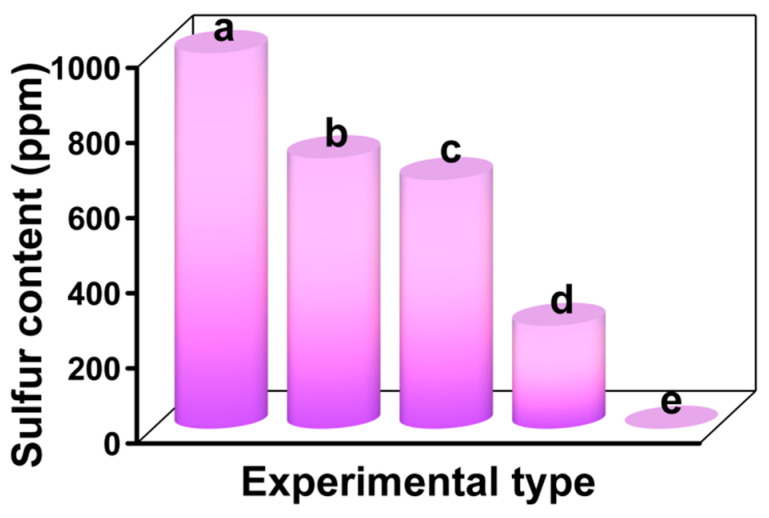
Effects of different desulfurization systems on sulfur removal: sulfur content before the reaction (a), extractive desulfurization of pure [BMIM]BF_4_ (b), unmodified [BMIM]Cl-based MCF in [BMIM]BF_4_ (c), ECODS of phosphomolybdic acid in [BMIM]BF_4_ (d), and ECODS of [BMIM]_3_PMo_12_O_40_-based MCF in [BMIM]BF_4_ (e). Experimental conditions: T = 60 °C, V_DBT_ = 5 mL, V_IL_ = 3 mL, O/S = 6, m([BMIM]_3_PMo_12_O_40_-based MCF) = 30 mg, and t = 3 h.

**Figure 13 materials-16-03195-f013:**
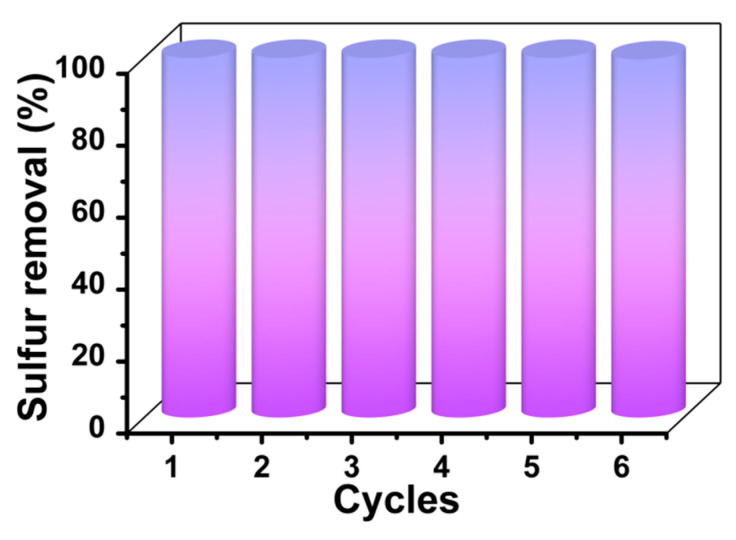
Effect of recycle runs of [BMIM]_3_PMo_12_O_40_-based MCF on sulfur removal efficiency. Experimental conditions: T = 60 °C, V_DBT_ = 5 mL, V_IL_ = 3 mL, O/S = 6, m([BMIM]_3_PMo_12_O_40_-based MCF) = 30 mg, and t = 3 h.

## Data Availability

Not applicable.

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
