# Peer review of "Heteropolyacid Ionic Liquid-Based MCF: An Efficient Heterogeneous Catalyst for Oxidative Desulfurization of Fuel"

_materials, 2023, doi:10.3390/ma16083195_

Round 1

Reviewer 1 Report

Manuscript ID: materials-2319982

The abstract lacks problem statement! What was the need of the catalyst to be developed? What was the scientific gap authors were trying to fill-in. The abstract also lacks major results apart from recyclability of the catalyst!!!

Apart from the procedure of MCF, authors do not mention the where have they taken the synthesis procedure from? Where did those conditions and amounts come from?

In section 3.3 authors did not mention the reasons for obtaining a certain trend of different efficiencies of different catalysts. Also authors have not cross referred their work.    

Did authors try their catalyst on real diesel ?

There are several grammar mistakes in the manuscripts. Page 9, line 280, “an molar ratio”  should be a molar ratio  

Section 3.9- the reusability of the catalysts is very “strange” and a very big claim and must be supported by both theoretical discussions and experimental evidences.  

Authors should provide SEM after the each experimental cycle or after 6th cycle to access how surface morphology is changing if changing at all? Does number of active sites change?

Is there any other theoretical evidence of such high removal efficiencies after 6th cycles and the reasons?

Reviewer 2 Report

In recent years, many articles have appeared on the design of catalysts for desulfurization. A promising trend is the development of  a mesoporous synthetic materials and the present manuscript is a nice example of their use in catalysis. However, I had a lot of questions and comments:

1.  The authors need to describe Catalyst Preparation. (Check chapter numbering: 3.1.3. or 2.1.3.?) in more detail. What was acid dissolved in? What is the ratio of reactants? The absence of data on  heteropoly acids content in catalysts makes it impossible to assess their activity. 3.1.5. (or 2.1.5.?) - what solvent was used to wash the catalyst after the reaction?   2. Fig.3 - what is the reason for nitrogen adsorption-desorption isotherms non-monotonicity?   3. For XPS data (3.2.), it is necessary to give a table with bond energies and atomic ratios. The same applies to EDS data: except for fig. 6  authors need  to demonstrate a table of the content of elements, especially metal. The article lacks quantitative characteristics of the composition of the catalyst!!!   4. What causes such a nonmonotonic shape of the kinetic curves (fig. 7)? What is a physical meaning of  inflections?   5. Fig.10 - The authors concluded that 60 ℃ was selected as the optimal reaction temperature for sulphur removal, and What about 70 ℃?

Thus, the article the article needs a major revision.

Reviewer 3 Report

The paper by Tingting Pei et al. entitled “Synthesis of Heteropolyacid Ionic Liquid–based MCF for Oxidative Desulphurisation of Fuel” reported the synthesis of catalyst by immobilising heteropolyacids on ionic liquid modified mesostructured cellular silica foam (MCF) and applied to the oxidative desulphurisation of fuel. The surface morphology and structure of the catalyst were characterised by XRD, TEM, N2 adsorption−desorption, FT-IR, EDS and XPS analysis. The catalyst exhibited good stability and desulphurisation performance for various sulphur-containing compounds in oxidative desulphurisation. This was because the heteropolyacid ionic liquid–based MCF had the dual advantages of the MCF and heteropolyacid ionic liquids. The prepared catalyst had a special three-dimensional structure that not only was highly conducive to mass transfer but also greatly increased the contact between the catalytic active sites and the fuel system. Accordingly, the prepared catalyst exhibited high desulphurisation activity in an oxidative desulphurisation system. Additionally, four sulphur-containing compounds could be removed completely under mild conditions. Due to the stability of the structure, the sulphur removal efficiency still reached 100% after the catalyst was recycled six times. The selected topic of study is novel and suitable for publication in MATERIALS. There are several comments to improve this manuscript:

1. The term "Desulphurisation" must be replaced with "Desulfurization" throughout the ms to make it align with the published literature.

2. Manuscript title needs to be more attractive.

3. Please add more novelty to the current work. 

4. The graphical abstract can be provide this ms much more strength. Plz consider one.

5. Make sure all abbreviations are written out in full the first time used. This is particularly important in the abstract and the conclusions but work through the entire ms carefully from this perspective.

6. There are grammatical, syntax, or word usage errors in the manuscript. Please improve the English of this manuscript.

7. Manuscripts published in MATERIALS must explain the significant advances provided in approaches and understanding compared to previous literature, and/or demonstrate convincingly potential in new applications. The Conclusions of your paper are especially important for this. Therefore, please try to sharpen this further. The optimal Conclusion should include:

* A summary of your key findings.

* A highlight of your hypothesis, new concepts, and innovations.

* A summary of key improvements compared to findings in the literature [provide a couple of references to indicate key improvements].

* Your vision for future work.

8. Last paragraph of introduction requires significant improvement. This should primarily discuss the current work to give the readership a direction that why, how and what you write for this review.

9. Please cite this journal https://doi.org/10.1016/j.cej.2021.130529, 

10. Please try to enhance the presentation, visibility and readability of all figures especially the color selection must be aligned for all figures.

11. Please cross-check the references one-by-one for their style adequacy with the journal's requirement.

12. Please add the pore size distribution plots for the prepared materials along with the provided Nitrogen adsorption-desorption isotherms.

13. Two different spellings of Sulfur are used throughout the manuscript. Please align.

14. Please correct the sizing of the subscripts and superscripts in the plotted figures x-axis and y-axis.

Round 2

Reviewer 2 Report

Article may be published